# An Antibacterial Peptide with High Resistance to Trypsin Obtained by Substituting d-Amino Acids for Trypsin Cleavage Sites

**DOI:** 10.3390/antibiotics10121465

**Published:** 2021-11-28

**Authors:** Xiaoou Zhao, Mengna Zhang, Inam Muhammad, Qi Cui, Haipeng Zhang, Yu Jia, Qijun Xu, Lingcong Kong, Hongxia Ma

**Affiliations:** 1College of Animal Medicine, Jilin Agricultural University, Changchun 130118, China; zhaoxo4466@163.com (X.Z.); zhangmn2021@126.com (M.Z.); dr.inam@sbbu.edu.pk (I.M.); cuiqi19941003@126.com (Q.C.); QJXu1997@163.com (Q.X.); 2Department of Animal Sciences, Shaheed Benazir Bhutto University, Sheringal, Dir Upper 18050, Pakistan; 3College of Life Science, Jilin Agricultural University, Changchun 130118, China; zhanghaipeng211@163.com (H.Z.); jiayu@jlau.edu.cn (Y.J.); 4The Engineering Research Center of Bioreactor and Pharmaceutical Development, Ministry of Education, Jilin Agricultural University, Changchun 130118, China

**Keywords:** antimicrobial peptides, trypsin, d-amino acids, stability, mechanism

## Abstract

The poor stability of antibacterial peptide to protease limits its clinical application. Among these limitations, trypsin mainly exists in digestive tract, which is an insurmountable obstacle to orally delivered peptides. OM19R is a random curly polyproline cationic antimicrobial peptide, which has high antibacterial activity against some gram-negative bacteria, but its stability against pancreatin is poor. According to the structure-activity relationship of OM19R, all cationic amino acid residues (l-arginine and l-lysine) at the trypsin cleavage sites were replaced with corresponding d-amino acid residues to obtain the designed peptide OM19D, which not only maintained its antibacterial activity but also enhanced the stability of trypsin. Proceeding high concentrations of trypsin and long-time (such as 10 mg/mL, 8 h) treatment, it still had high antibacterial activity (MIC = 16–32 µg/mL). In addition, OM19D also showed high stability to serum, plasma and other environmental factors. It is similar to its parent peptide in secondary structure and mechanism of action. Therefore, this strategy is beneficial to improve the protease stability of antibacterial peptides.

## 1. Introduction

In recent decades, bacterial resistance has become more and more serious due to the abuse of antibiotics [1,2]. The emergence of new antibiotic-resistant bacteria means that clinical treatment is facing a great crisis. Therefore, the development of new antibacterial agents has become an urgent task [3]. Antimicrobial peptides (AMPs) are considered ideal candidates to replace antibiotics because of their strong potential against multidrug-resistant bacteria and their unique mechanism of action; that is these peptides are currently the main focus of research [3,4,5]. At present, most of the studies on antimicrobial peptides have focused on finding new peptides with good activity, but few studies have focused on their behavior in the clinical environment [6,7]. However, there are a variety of proteases in organisms, such as trypsin, which can lead to the degradation and inactivation of antimicrobial peptides. These proteases are one of the barriers preventing the clinical application of antimicrobial peptides [6,8]. Although some newly designed and discovered antimicrobial peptides have also been studied on protease resistance, most of them were only studied for the stability and hydrolysis rate of antimicrobial peptides at low concentrations of protease (e.g., 0.1 mg/mL) [9,10]. Generally, few studies have been conducted on the stability and activity changes of antimicrobial peptides at high concentrations (e.g., 10 mg/mL). In fact, the protease concentration in vivo may be far more than 0.1 mg/mL. In recent years, Wang et al. designed an antimicrobial peptide resistant to high concentrations of protease and carried out related research [11].

Trypsin is an important protease in mammals. Like most proteases, trypsin poses a great threat to the application of antimicrobial peptides [12,13]. Trypsin preferentially lyses Lys and Arg cationic residues located at P1. However, these two residues are critical to the activity of AMPs [11,14]. This is because they can produce electrostatic interaction with some charged organelles of bacteria, which affects the normal life activities of bacteria. Therefore, Arginine and Lysine are not only the cleavage sites of an enzyme, but also affect antibacterial activity [10,15,16,17]. Therefore, a question arises: how can we retain the activities of AMPs and improve their stability to trypsin? In order to improve the stability of antimicrobial peptides, many strategies have been proposed, such as cyclization [18], peptidomimetic [19], N-methylation [20,21], side-chain modification [17], multimerization [22], drug delivery methods [23] and the incorporation of unnatural amino acid substitutions [24]. In comparison to the above methods, d-amino acid substitution is a simple and inexpensive strategy to improve antimicrobial peptide–proteinase resistance [9]. d-amino acids are enantiomers of l-amino acids and belong to a group of non-natural amino acids [25]. Their molecular weights, formulas and side chains are the same, except for their spatial conformations. d-type amino acids can retain the physicochemical properties of corresponding l-type amino acids to the greatest extent. The incorporation of d-type amino acids makes it difficult for protease to recognize sensitive amino acids, which can improve protease resistance [26,27]. For example, replacing the cleavage sites of *Pseudomonas aeruginosa* elastase and *Staphylococcus aureus* metalloproteinase in the LL-37 turned sequence with d-amino acid can effectively resist protease degradation [28].

In this study, we tried to use this strategy to increase the resistance of trypsin to antimicrobial peptides. OM19R is a polyproline cationic antimicrobial peptide designed by our team. It has high antibacterial activity against some gram-negative bacteria and low toxicity in vitro. It has high stability against some environmental factors such as physiological salt and serum, but its resistance to trypsin needs to be improved [29]. We replaced all l-Arginine and l-Lysine in OM19R with d-Arginine and d-Lysine to obtain a new peptide OM19D. It not only maintained all its advantages, including high antibacterial activity but also improved its resistance to trypsin, especially at high concentrations of trypsin. In addition, the antibacterial mechanism of OM19D was investigated, and we found that its target was not the cell membrane but intracellular, similar to its parent peptide OM19R.

## 2. Results

### 2.1. The Structure-Activity Relationship (SAR) of OM19R

The antimicrobial peptide OM19R, VDKPPYLPRPRPIRRPGGR-NH_2_, consisted of 19 l-amino acids, including 1 l-Lysine (Lys, K) and 5 l-Arginine (Arg, R). The effect of each amino acid residue on the antibacterial activity of OM19R was studied by the l-alanine (Ala, A) scanning method. As shown in Figure 1, the antibacterial activity of OM19R disappeared completely when the tyrosine at position six and the leucine at position seven were replaced, while the antibacterial activity of OM19R decreased only 0–8 folds when other sites were replaced. This suggests that amino acids at positions six and seven were important and irreplaceable in the whole structure.

### 2.2. Peptide Design and Antibacterial Activity

All l-Arginine and l-Lysine in OM19R were replaced by d-Arginine and d-Lysine to obtain a new peptide named OM19D. The sequence and key physicochemical parameters of the peptides are shown in Table 1. According to the results, the molecular weight and charge number of OM19R and OM19D were the same. The measured molecular weights of all peptides were consistent with the theoretical values, which prove that they are synthesized. The ESI-MS mass spectrometry results are presented in Appendix A. Their purity is over 98%, and the HPLC chromatographic profiles of the peptides are shown in the supplementary diagram of Appendix A.

### 2.3. Circular Dichroism (CD) Spectroscopy

The secondary structures of OM19D and OM19R were detected by CD spectroscopy in phosphate buffer (10 mM, PBS), trifluoroethanol (50% *v*/*v*, TFE) and sodium dodecyl sulfate (30 mM, SDS). A 50% TFE solution simulates the hydrophobic environment of the bacterial cell membrane, while SDS solution simulates the negatively charged environment of the prokaryotic cell membrane [30]. Both of them can increase the helicity of peptides. As shown in Figure 2, the secondary structure patterns of the two peptides in the three environments were consistent with the characteristics of random coiling (strong negative peak below 200 nm and a small and wide positive peak around 220 nm). In comparison with the parent peptide OM19R, the secondary structure of the designed peptide OM19D did not change significantly.

### 2.4. Antibacterial Activity

As shown in Table 2, OM19D is similar to its parent peptide OM19R in its antimicrobial spectrum and antimicrobial activity. For standard strains, the MIC values of OM19D were 1–4 folds lower than OM19R. For the clinically isolated drug-resistant strains, some strains were slightly sensitive to OM19D, and some were slightly sensitive to OM19R. These drug-resistant strains were resistant to more than three antibiotics with different mechanisms, and their MICs for antibiotics were shown in Appendix A.

### 2.5. Stability toward Tryptic Degradation

The results of the antibacterial activity of peptides treated with trypsin at different concentrations and different times are shown in Table 3. In short, OM19D showed high antibacterial activity (MIC = 16 µg/mL), even at high concentration of trypsin (10 mg/mL), while the parent peptide OM19R and control melittin had lost their antibacterial activity at 5 mg/mL trypsin (MIC > 128 µg/mL). When compared with untreated, the MIC reduced by 8–16 folds. Moreover, OM19D still had high antibacterial activity (16–32 µg/mL) after being treated with 10 mg/mL trypsin for 8 h, while OM19R and melittin had already lost their antibacterial activity. Therefore, the designed peptide OM19D was resistant to long-term treatment with a high concentration of trypsin.

### 2.6. Condition Sensitivity Assays

As shown in Table 4, Table 5 and Table 6, OM19D still has high antibacterial activity in different physiological salt environments, high-temperature environments and acid-base environments, which is not much different from before treatment. In addition, OM19D still maintained antibacterial activity in serum and plasma. Parent peptide OM19R and melittin were the control groups.

### 2.7. In Vitro Safety Evaluation

The safety was evaluated by hemolysis rate and cytotoxicity in vitro. The hemolytic activity and cytotoxicity of peptides were measured in the range of 1–512 µg/mL. As shown in Figure 3, compared with the control group melittin, even when the concentration is higher than the minimum inhibitory concentration, the hemolysis rate of rabbit red blood cells caused by OM19D is lower than 10%, and the survival rate of mouse macrophages (RAW 264.7) can reach more than 80%.

### 2.8. Antimicrobial Mechanism Study

#### 2.8.1. Growth Curves of Bacteria and Time-Kill Curve

As shown in Figure 4, according to the growth curve, OM19D can completely inhibit the growth of bacteria when the concentration is greater than or equal to MIC. According to the bactericidal kinetics curve, almost all bacteria could be killed by OM19D within 8 h, and the bactericidal rate was proportional to the concentration.

#### 2.8.2. AKP Activity Determination

AKP (alkaline phosphatase) is found between bacterial cell walls and cell membranes [31]. When the cell wall was intact, its activity was almost undetectable in the bacterial fluid. Once the cell wall is damaged, AKP leaks into the bacterial fluid, where it can be detected. Therefore, the integrity of the cell wall can be reflected by the change of AKP content in bacterial fluid. As shown in Figure 5, there was no significant outflow of AKP after 4 h of the peptide treatment compared with negative control group PBS. This shows that the cell integrity has not been destroyed.

#### 2.8.3. Outer Membrane and Inner Membrane Permeability Assay

As shown in Figure 6, the effects of OM19D and OM19R on the permeability of the bacterial outer membrane and inner membrane were compared by fluorescence intensity analysis. Compared with PBS in the negative control group, there was no obvious change in fluorescence intensity in each group. However, when compared with the positive control group (outer membrane: polymyxin; inner membrane: Triton X-100), the fluorescence intensity of each group had significant changes. Therefore, the designed peptide OM19D was similar to the parent peptide OM19R and had no significant effect on the permeability of the bacterial outer membrane and inner membrane.

#### 2.8.4. Proton Motive Force

Proton motive force (PMF) is composed of the pH gradient (ΔpH) and the potential gradient (Δφ), which together constitute the electrochemical gradient(ΔP) [32]. Fluorescent probe disc_3_(5) was used to detect membrane potential, and fluorescent probe BCECF-AM was used to detect intracellular pH changes. As shown in Figure 7, when compared with the negative control group PBS, the membrane potential and pH of OM19D and OM19R did not change significantly. This suggests that the designed peptide OM19D was similar to the parent peptide OM19R, as they did not disrupt the PMF of the bacteria.

#### 2.8.5. Intracellular ATP

As shown in Figure 8, the intracellular ATP of bacteria treated with OM19D decreased significantly and showed a concentration-dependent decrease. Likewise, OM19R showed a similar trend.

#### 2.8.6. Total ROS Level

The intracellular ROS of bacteria treated with the peptides was detected by fluorescence probe DCFH-DA. As shown in Figure 9, the levels of ROS in OM19D treated bacteria were significantly higher than those in untreated bacteria, which was similar to the parental peptide OM19R.

#### 2.8.7. DNA Binding Assay

As shown in Figure 10A,B, the designed peptide OM19D showed the ability to bind to genomic DNA at a concentration of 512 μg/mL, and the parent peptide OM19R showed the ability to bind to DNA at a concentration greater than or equal to 256 μg/mL Their concentration of binding to DNA was much higher than the MIC. As shown in Figure 10C,D, OM19D and OM19R showed the ability to bind to small DNA fragments at the concentration of 16 μg/mL. Therefore, OM19D was similar to OM19R, and interference with bacterial genetic material was not its main mechanism of action.

#### 2.8.8. Quantification of Intracellular Protein Concentration

As shown in Figure 11, OM19D treatment caused bacterial protein concentration reduction in a time-dependent and a dose-dependent manner, respectively. The parental peptide OM19R also showed a similar trend. Therefore, OM19D showed the ability to affect the synthesis of bacterial proteins, as did the parent peptide OM19R.

#### 2.8.9. Non-Specific Efflux Pumps

The effects of the peptides on the non-specific bacterial pumping system were detected by fluorescence probe Ethidium bromide (EtBr), and the EtBr efflux from cells was monitored for 120 min. As shown in Figure 12, compared with the control group (decoupling agent CCCP), EtBr efflux from bacteria treated with OM19D showed a decreasing trend in a time-dependent manner, and so did OM19R. The results showed that OM19D and OM19R had no inhibitory effect on the bacterial efflux pump.

## 3. Discussion

Up to now, more than 4500 AMPs have been stored in the antimicrobial peptide database (APD), and the number is still increasing [33,34,35]. However, there are only a few available for clinical use, such as bacitracin and polymyxin [36]. Antimicrobial peptides are easily degraded by protease and lose their antibacterial activity, which leads to the limitations of clinical application [3]. Among them, trypsin is a very important digestive enzyme ubiquitous in mammals, which mainly attacks the arginine and lysine of proteins or polypeptides [37]. In this study, we focused on trypsin and focused on overcoming its degradation of antibacterial peptides. According to our previous reports, OM19R has high activity against some intestinal pathogenic bacteria such as *Escherichia coli* and *Salmonella* [29]. Therefore, it is likely to be used as an intestinal microbicide in the future. Therefore, trypsin is one of the obstacles that must be overcome.

According to the structure–activity relationship of OM19R, the cleavage site of trypsin, and the structural characteristics of l-Arginine and l-Lysine, we chose the strategy of replacing l-amino acid at the cleavage site with d-amino acid, in other words, replacing all l-Arginine and l-Lysine in OM19R with d-Arginine and d-Lysine, and obtaining the designed peptide OM19D. According to the results, OM19D still maintained high activity (MIC = 16 μg/mL) after being treated with a high concentration of trypsin (10 mg/mL). When compared with the untreated, MIC decreased by only 8–16 folds. The reason for the slight loss of antibacterial activity of OM19D may be that a very small fraction of peptide is degraded during the operation of inactivating trypsin. In addition, there is a potential possibility that a fraction of peptide is co-precipitated with trypsin that resulted in an increased MIC. The trypsin concentration of 10 mg/mL was chosen because it was consistent with the simulated intestinal fluid (SIF) content specified in the United States Pharmacopoeia [38]. However, the parent peptide OM19R and the control group melittin have lost their antibacterial activity. In addition, OM19D still maintained its antibacterial activity after being treated with 10 mg/mL trypsin for 8 h. Therefore, it can be concluded that the strategy not only improves the resistance of the peptide to trypsin but also maintains its antibacterial activity. Importantly, it can resist high concentrations of trypsin for a long time.

d-type amino acid substitution strategy has the advantages of simplicity, low price and relatively low biological toxicity [3]. Polymyxin and daptomycin, which have been put on the market, all contain d-type amino acids in their structures [39,40]. In this study, l-Lysine and d-Lysine, and l-Arginine and d-Arginine, have the same molecular weight, molecular formula, primary amino (Lysine) and guanidine (Arginine) in the side chains, which play a critical role in antibacterial activity, but they are only different in spatial conformation. Therefore, the antibacterial activity of the peptide was almost unaffected by d-amino acid substitution; however, resistance to trypsin was significantly increased. According to the results of this experiment, the designed peptide OM19D is completely in line with the original expectations, which not only retained the antibacterial activity but also enhanced the stability of trypsin. This shows the feasibility of this strategy.

The stability of antimicrobial peptides in the blood is also very important, as no matter the method of administration, the drug will eventually be absorbed into the blood. We tested the stability of OM19D in bovine serum and rabbit plasma. After treatment, the peptide maintained antibacterial activity. In addition, we also tested that it can still have high antibacterial activity in the physiological salt environment, extreme temperatures and an acid-base environment. OM19D exhibits heat-cold and acid-base stability, which is important for many production and storage processes such as medicine, food or feed processing [35]. For in vitro safety, we evaluated its hemolytic activity and cytotoxicity. According to previous reports, and the results of this experiment, the cytotoxicity of both the designed and the parent peptide OM19R are low. In comparison to OM19R, the designed peptide OM19D has no increase in hemolytic activity and cytotoxicity even if it contains unnatural amino acids, which is within the reasonable range.

Finally, the action mechanism of OM19D was preliminarily discussed. The main targets of antimicrobial peptides include external targets that target the cell membrane (e.g., Melittin) or cell wall (e.g., Nisin), and intracellular targets such as nucleic acid synthesis (e.g., Buforin II), protein synthesis (e.g., Indolicidin), inhibition of efflux pump (e.g., PAβN) and ATP efflux (e.g., Histatins), etc. [41,42,43]. According to the target of antimicrobial peptides, the following studies were performed. Circular Dichroism showed that OM19D and OM19R had similar secondary structures, which correlated with the characteristics of a random coil. Therefore, the incorporation of d-amino acids did not cause significant changes in the secondary structure. According to previous reports, the mechanism of OM19R is not to destroy the bacterial cell membrane. We have also done related research and found that OM19D does not destroy the integrity of cells and does not affect the permeability of the cell membrane, which is similar to the parent peptide OM19R. This is similar to most polyproline antimicrobial peptides. The targets of the polyproline antimicrobial peptides are generally inside cells. Therefore, we did some intracellular research. We found that the bacteria treated with OM19D had no significant effect on the proton-powered and nonspecific effluent systems. It only binds to DNA when it is 8 folds higher than MIC. However, the intracellular reactive oxygen species increased significantly, which may be due to the entry of polypeptides into cells affecting their normal life activities, resulting in stress response or its normal redox response being affected. Intracellular ATP concentration decreased significantly, which may destroy the energy chain. The decrease of ATP may also affect protein synthesis. The results showed that OM19D had a significant effect on intracellular protein concentration, and the protein concentration decreased significantly with the increase of time and concentration. In all the experiments related to the antibacterial mechanism of OM19D, we used the parent peptide OM19R as the control group. We found that their antibacterial mechanisms are very similar. The main mechanism of OM19D action is to affect intracellular activity rather than to destroy cell integrity, as most antimicrobial peptides do. Therefore, the incorporation of d-amino acids did not affect the antibacterial mechanism.

## 4. Materials and Methods

### 4.1. Materials

#### 4.1.1. Bacterial Species

Standard strains: *Escherichia coli* ATCC25922, *Escherichia coli* k88, *Salmonella enterica* ATCC 14028, *Salmonella typhimurium* CMCC50115, *Shigella flexneri* ATCC12022, *Shigella dysenteriae* CMCC51252, *Klebsiella pneumoniae* ATCC25955, *Staphylococcus aureus* ATCC25923. Clinically isolated strains: *Escherichia coli* SN5, *Escherichia coli* S1N1, *Escherichia coli* w136, *Escherichia coli* w122, *Escherichia coli* w123, *Escherichia coli* w124, *Escherichia coli* QY, *Shigella flexneri* QY1, Methicillin-resistant *Staphylococcus aureus* HP. RAW 264.7 cells. All strains were obtained from the College of Animal Science and Technology, Jilin Agricultural University (Changchun, China).

#### 4.1.2. Reagent

Mueller–Hinton broth (MHB) and Mueller–Hinton agar (MHA) were purchased from GL Biochem (Shanghai, China). Triton X-100 and phosphate-buffered saline (PBS) solution were purchased from Macklin (Shanghai, China). BCECF-AM, EtBr, PI, diSC_3_(5), polymyxin B, 4-(2-hydroxyethyl) piperazine-1-ethanesulfonic acid (HEPES) and NPN were also purchased from Macklin (Shanghai, China). TransDetect Cell Counting Kit-8 (CCK-8) and Dulbecco’s modified Eagle’s medium with high glucose (DMEM) were purchased from TransGen Biotech (Beijing, China). Fetal bovine serum (FBS) was purchased from Gibco (Shanghai, China).

### 4.2. Synthesis and Sequence Analysis of Peptides

The alanine scan was performed by replacing each amino acid residue in the original peptide OM19R with alanine. OM19D was the replacement of l-arginine and l-lysine in the original peptide OM19R with d-arginine and d-lysine. The designed peptides were synthesized and purified by GL Biochem (Shanghai, China). The peptides were synthesized using standard Fmoc protocols. The Peptides were purified and determined by RP-HPLC with SHIMADZU Inertsil ODS-SP (4.6 mm × 250 mm × 5 μm, RP-C18) column, UV absorption wavelength 214 nm, volume 60 μL, eluted with water/acetonitrile gradient containing 0.1% trifluoroacetic acid, flow rate 1.0 mL/min. Finally, ESI-MS was used to determine the molecular weight.

### 4.3. MIC Measurements

Minimum inhibitory concentrations (MICs) of OM19R and its derivatives were determined by the broth micro-dilution method, which was in accordance with CLSI 2015 guidelines [44]. Briefly, bacteria were cultured in Mueller–Hilton broth (MHB) medium to logarithmic phase and diluted to 10^5^ CFU/mL. Then, 50 μL of peptides of different concentrations and 50 μL bacterial solution were mixed in a sterile 96-well plate, and the plates were incubated for 16 h at 37 °C. MIC was examined by measuring the OD at 600 nm. The MIC was defined as the lowest peptide concentration that inhibited 95% of the bacterial growth.

### 4.4. CD Measurements

The test methods refer to previous reports [5]. The CD spectra of the peptides were detected on a J-810 spectropolarimeter (Jasco, Tokyo, Japan). The peptides were respectively dissolved in 10 mM PBS (pH 7.4), 50% TFE, and 30 mM SDS where the final concentration was 128 µM. The experimental parameters were as follows: 10 nm·min^−1^ scanning rate, 1 mm path length, 190–260 nm wavelength range, and 25 °C. Each sample was scanned at least three times.

### 4.5. Typsin Resistance Assays

The effect of different concentrations of trypsin on the antimicrobial activity of peptides was tested using previously reported methods [11]. Trypsin was prepared to the required concentration. In the sterile 96-well plates, the peptide was continuously twofold diluted to different concentrations in trypsin solution. The plates were incubated at 37 °C, and the incubation time was determined according to the experiment requirements. The peptide alone and protease alone were used as a control. Samples were taken at different times, heated at 100 °C for 15 min to inactivate the protease, and centrifuged at 13,000× *g* for 30 min to precipitate the protease. Finally, an equal volume of bacteria (10^5^ CFU/mL) was added to the 96-well plates, the plates were incubated for 16 h at 37 °C, and then the MIC value of the peptide was observed. The indicator bacterium was *Escherichia coli* ATCC25922. This experiment was repeated three times.

### 4.6. Physiological Salt Sensitivity Assays

The peptide salt sensitivity was evaluated by measuring the MIC. The method is as previously described [11,45]. The peptides were diluted in deionized water containing different salts, and then an equal volume of *Escherichia coli* ATCC25922 solution (10^5^ cfu/mL) was added. The mixture was incubated at 37 °C for 16 h. The final physiological concentrations of these salts were 150 mM NaCl, 4.5 mM KCl, 6 μM NH_4_Cl, 1 mM MgCl_2_, 8 μM ZnCl_2_ and 4 μM FeCl_3_, respectively. This experiment was repeated three times.

### 4.7. Temperature and pH Sensitivity Assays

The temperature and pH sensitivities of the peptides were determined by an MIC assay as described previously [30]. For temperature sensitivity, the diluted peptide solution was incubated at 0 °C, 37 °C, 42 °C, 65 °C and 100 °C, respectively, for 30 min, then stopped heating and returned to room temperature. An equal volume of *Escherichia coli* ATCC25922 (10^5^ CFU/mL) was added, and the mixture was incubated at 37 °C for 16 h. MIC values were observed. This experiment was repeated three times.

For pH sensitivity, we adjusted the diluent to pH 4.0, 6.0, 8.0 and 10.0 with HCl or NaOH, and then diluted the peptide with this diluent. The peptide solutions of different pH were incubated for 1 h, and then the pH of the reaction solution was neutralized by HCl or NaOH to terminate the reaction. An equal volume of *Escherichia coli* ATCC25922 (10^5^ CFU/mL) was added, and the mixture was incubated at 37 °C for 16 h. MIC values were observed. This experiment was repeated three times.

### 4.8. Serum and Plasma Sensitivity Assays

The serum sensitivities of the peptides were determined by an MIC assay as described previously [11,30]. Bovine serum and rabbit plasma solutions of different concentrations (12.5%, 25%, 50% and 100%) were used as the diluent of the peptide, and the peptides were twice diluted and incubated at 37 °C for 1 h. After that, the reaction was terminated after treatment at 100 °C for 15 min and cooled to room temperature. The same volume of *Escherichia coli* ATCC25922 solution (10^5^ CFU/mL) was incubated for 16 h at 37 °C. The change of MIC value was observed. This experiment was repeated three times.

### 4.9. Measurement of Hemolysis Activity

We used fresh rabbit red blood cells and sheep blood red blood cells to test the hemolytic activity of the peptide, as previously reported [10]. The collected red blood cells were washed three times with sterile 0.9% NaCl solution (pH 7.4) and diluted to 2% (*v*/*v*). The peptides were two-fold diluted with 0.9% NaCl solution to different concentrations. The equal volume of red blood cell suspension and different concentrations of peptide solution was added to the 96-well plate and incubated at 37 °C for 1 h. Then, the mixture was centrifuged at 1000× *g* for 10 min at 4 °C, the supernatant was transferred to a new 96-well plate, and the absorbance was measured at OD 570 nm using a microplate reader (Microplate reader, TECAN GENios F129004, Tecan, Salzburg, Austria). A 0.9% NaCl erythrocyte suspension without peptide was used as negative control and 1% Triton X-100 as a positive control. The hemolysis rate was calculated according to the following equation:Hemolysis rate (%) = [(OD_576 sample_ − OD_576 blank_)/(OD_576 1%Triton X-100_ − OD_576 blank_)] × 100%(1)

### 4.10. Cytotoxicity Assays

The CCK-8 assay was used to detect the cytotoxicity of the peptide, as previously reported [30,31]. RAW 264.7 cells were cultured in DMEM (containing 10% fetal bovine serum), and 10,000 cells were added to each well in 96-well plates and incubated at 37 °C in 5% CO_2_ for 6 h. The peptide was two-fold diluted in DMSO and added to 96-well plates in equal volume. The mixture was cultured for 16 h. Then, CCK-8 (10%, *v*/*v*) was added to the cell culture medium to each well and cultured for 4 h. The absorption was measured at OD 450 nm using a microplate reader. The blank medium was used as the negative control, and the medium containing only cells was used as the positive control. Cell survival rate was calculated according to the following equation:Cell survival rate (%) = [(OD_450 sample_ − OD_450 negative control_)/(OD_450 positive control_ − OD_450 negative control_)] × 100%(2)

### 4.11. Determination of AKP Outflow

The test methods refer to previous reports [46]. AKP of *Escherichia coli* ATCC 25922 were determined using an AKP Assay Kit (Jiancheng, Nanjing, China). Collected cells in the logarithmic growth phase were rinsed with 0.01M PBS (pH7.4) three times and diluted to OD600 = 0.5. Peptides of different concentrations were added and incubated for 4 h and then centrifuged at a low temperature. The PBS treatment group was a negative control group, and Triton X-100 treatment group was a positive control group. The supernatant was collected and placed in the working solution, following the instructions, and the absorbance was measured at OD520 nm. Then, the value was substituted into the formula to find the AKP concentration.

### 4.12. Outer Membrane Permeability Assay

Fluorescent dye NPN was used to examine outer membrane permeability according to previously reported methods [47]. Collected bacterial cells in the logarithmic growth phase were rinsed with 5mMHEPES buffer (pH7.4, contains 5 mM glucose) three times and diluted to OD600 = 0.5. Then, a 10 μM NPN was added to the bacterial suspension and incubated at room temperature for 30 min. An equal volume of different concentrations of peptides and the bacterial suspension was added to the black 96-well plate. The fluorescence intensity of the samples was measured by F4500 fluorescence spectrophotometer (emission λ = 420 nm, excitation λ = 350 nm). The negative control was the initial fluorescence value of *Escherichia coli* ATCC25922 bacterial suspension containing NPN, and the positive control was the fluorescence value produced by polymyxin B on *Escherichia coli* ATCC25922 cells.

### 4.13. Inner Membrane Permeability Assay

Fluorescent dye PI was used to examine outer membrane permeability according to previously reported methods [48]. The collected bacterial cells in the logarithmic growth phase were rinsed with 0.01 M PBS buffer (pH 7.4) three times and diluted to OD600 = 0.5. Then, a 10 μM PI was added to the bacterial suspension and incubated at room temperature for 10 min. The suspension was placed in each well of the 96-well plate and cultured at 37 °C with peptides ranging from 0.5 × MIC to 4 × MIC. The fluorescence intensity of the samples was measured by the F4500 fluorescence spectrophotometer (emission λ = 617 nm, excitation λ = 535 nm). The negative control was the initial fluorescence value of *Escherichia coli* ATCC25922 bacterial suspension containing PI, and the positive control was the fluorescence value produced by Triton X-100 on *Escherichia coli* ATCC25922 cells.

### 4.14. Proton Motive Force Assay

#### 4.14.1. Membrane Potential Assay

The test methods refer to previous reports [48]. *Escherichia coli* ATCC25922 cells were washed and resuspended with 5 mM HEPES (pH 7.0, plus 5 mM Glucose). The potentiometric probe DiSC_3_(5) was added in the medium at a final concentration of 1 × 10^−6^ M. Peptides of different concentrations were added to the culture medium containing bacteria and DISC_3_(5), and incubated in the dark for 30 min. The excitation wavelength at 622 nm and emission wavelength at 670 nm, measured fluorescence intensity.

#### 4.14.2. Intracellular pH Assay

The test methods refer to previous reports [48]. *Escherichia coli* ATCC25922 cells were washed and resuspended with 5 mM HEPES (pH 7.0). Fluorescent probe BCECF-AM was added to the medium at a final concentration of 20 μM. After the fluorescence stabilized, glucose (25 μM) or varying peptides were added. The excitation wavelength and emission wavelength were set at 488 nm and 522 nm, respectively.

### 4.15. Intracellular ATP Determination

Intracellular ATP levels of *Escherichia coli* ATCC25922 were determined using an ATP Assay Kit (Jiancheng, Nanjing, China). Collected cells in the logarithmic growth phase were rinsed with 0.01 M PBS (pH7.4) three times and diluted to OD600 = 0.05. Peptides of different concentrations were added and incubated for 1 h, then centrifuged at a low temperature. The supernatant was collected and placed in the ATP working solution, mixed, stood for 5 min, and the absorbance was measured at 636 nm. Then, the value was substituted into the formula to find the ATP concentration [48].

### 4.16. Total ROS Measurement

Total ROS levels of *Escherichia coli* ATCC 25922 were determined using a ROS Assay Kit (Jiancheng, Nanjing, China). Collected cells in the logarithmic growth phase were rinsed with 0.01 M PBS (pH7.4) three times and diluted to OD600 = 0.5. DCFH-DA dye (final concentration: 10 μM) was added into the bacterial suspension and incubated for 30 min at room temperature. Then, the cells were washed with PBS to remove fluorescent probes that had not entered the cells. After that, different concentrations of antimicrobial peptides were added and let stand for 1 h at 37 °C. Subsequently, the fluorescence intensity was immediately measured with the excitation wavelength at 488 nm and emission wavelength at 525 nm. Hydrogen peroxide as the hydrogen donor was used as a positive control of ROS production [49].

### 4.17. Efflux Pump Assay

Cells were co-incubated with 5μM EtBr and sub-MIC of peptides or known efflux pump inhibitor CCCP (100 μM) at 37 °C to OD600 = 0.5. After centrifuging at 8000× *g* at 4 °C for 10 min, the precipitate was collected and resuspended in MHB. Then, EtBr effluent was monitored within 120 min at an excitation wavelength of 525 nm and an emission wavelength of 605 nm [44,48].

### 4.18. Quantification of Intracellular Protein

#### 4.18.1. Test of Protein Concentration Change with Peptides Concentration Change

A 5mL *Escherichia coli* ATCC25922 was cultured to the logarithmic growth stage. Bacteria were treated with the final concentration of 0.5 × MIC to 4 × MIC peptide and 0.01 M PBS (pH = 7.4) for 1 h. The bacteria were collected by centrifugation and washed with PBS. Then, 1mL PBS was added to resuspend the bacteria. The total number of bacteria at each concentration point was obtained by the flat colony counting method. Meanwhile, the bacterial resuspension was ultrasonically crushed in an ice bath for 5 min. The supernatant was collected and centrifuged at 4 °C for 30 min. Finally, the total protein concentration was determined using the BCA kit (Sangon, Shanghai, China) [49]. The bacterial survival rate of the PBS group was set as 100%, and the bacterial survival rate of the other concentration point was calculated. The actual total protein concentration, measured at each concentration point, was divided by the survival rate of each concentration point so as to get the relative total protein concentration when the survival rate of bacteria at each point was 100%.

#### 4.18.2. Test of Protein Concentration Change with Action Time

A 5mL *Escherichia coli* ATCC25922 was cultured to the logarithmic growth stage. Bacteria were treated with peptides whose final concentration was subinhibitory concentrations for 0, 1, 2, 4 and 8 h, respectively. The total number of bacteria at each time point was obtained by the flat colony counting method. The remaining methods were as described above. The bacterial survival rate of 0-h treatment group was set as 100%, and the bacterial survival rate of the other time point was calculated. The actual total protein concentration measured at each time point was divided by the survival rate of each time point so as to get the relative total protein concentration when the survival rate of bacteria at each point was 100%.

### 4.19. DNA Binding Assay

DNA combined experiments were based on previous reports [30,50]. Collected *Escherichia coli* ATCC25922 cells in the logarithmic growth phase were rinsed with PBS (pH 7.4) three times and diluted to OD600 = 0.5. Then, the genomic DNA was extracted, or Small DNA fragments were obtained by PCR amplification of 16SrDNA. The primer sequences are shown in Appendix A. A 400 ng genomic DNA or small DNA fragments was incubated with peptide solutions of different concentrations in the binding buffer at 37 °C for 2 h. After that, the samples were detected via 1% agarose gel electrophoresis.

### 4.20. Growth Curves of Bacteria and Time-Kill Curve

The bacterial growth curve and time-kill curve was measured as previously reported [51].

#### 4.20.1. The Bacterial Growth Curve

The single colony of *Escherichia coli* ATCC25922 was selected and inoculated in 1 mL MH liquid medium and cultured at 37 °C to the logarithmic growth phase. The culture was standardized to meet the 0.5McFarland turbidity standard. Then, the bacterial liquid was diluted in an MHB medium 1:100. Finally, different concentrations of antimicrobial peptides were added to the 96-well plate, and then diluted bacterial solution of the same volume was added. The absorption was measured at 600 nm using a microplate reader every 1 h for 24 h.

#### 4.20.2. Time-Kill Curve

*Escherichia coli* ATCC 25922 (10^5^ CFU/mL) was treated with peptides of 0.5 × MIC and 2 × MIC concentrations. A 50 μL sample was taken out every 10 min and diluted 100-fold, then coated on MHA solid medium and cultured at 37 °C. Colonies were counted at each sampling time point, and the time-kill curve was drawn. The samples without antimicrobial peptides were the control group.

### 4.21. Statistical Analysis

Statistical analysis was performed using unpaired *t*-test between two groups and one-way ANOVA among multiple groups. SPSS22.0 was used for statistical analysis. Quantitative data were expressed as mean ± standard deviation, *p*-values was considered statistically significant (* *p* <0.05,** *p* < 0.01, *** *p* < 0.001, ns represents insignificant).

## 5. Conclusions

In this study, according to the structure–activity relationship, we successfully designed a polypeptide with high resistance to trypsin by replacing arginine and lysine at the trypsin restriction site with d-type amino acid and tested its biological activity. Compared with the parent peptide OM19R, the antimicrobial activity of the designed peptide OM19D was not significantly decreased but the stability of trypsin was significantly improved, especially in a high concentration of protease. OM19D still had an antibacterial activity after being treated with a high concentration of trypsin for 8 h and showed strong resistance to trypsin. In addition, OM19D is well tolerated in serum and plasma and is resistant to high temperatures, acid and alkali. It has no obvious hemolytic activity and cytotoxicity. According to CD spectra, the designed peptide OM19D and the parent peptide OM19R conform to the irregular curl characteristic, indicating that part of d-type amino acid substitution did not affect its secondary structure. Finally, the target of the designed peptide OM19D is not on the cell membrane, while the intracellular target may exist as a potential possibility, which is similar to the parent peptide OM19R. These results suggest that the substitution of d-type amino acids for amino acids at the cleavage site of protease is helpful to improve the resistance of antimicrobial peptides to protease, which is beneficial to future clinical studies.

## Figures and Tables

**Figure 1 antibiotics-10-01465-f001:**
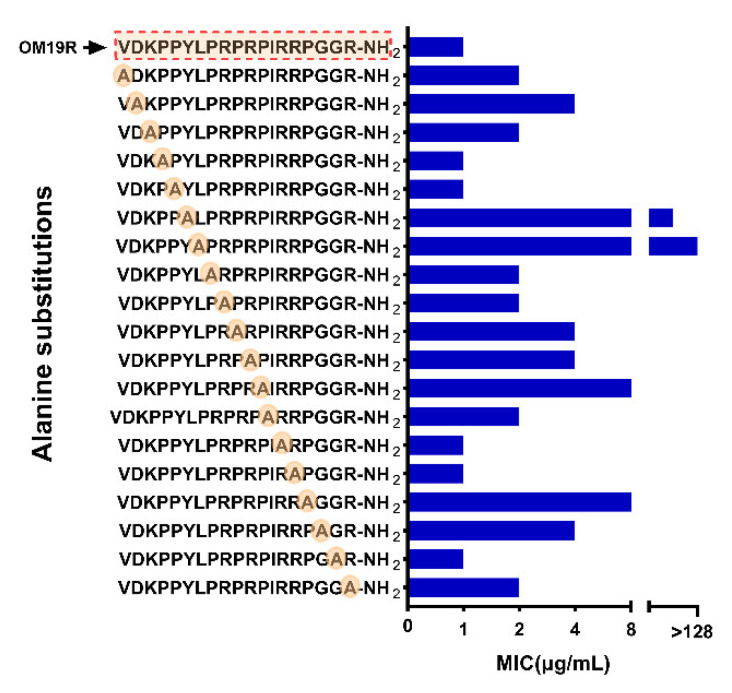
Antibacterial activity of l-Alanine scanner derivatives of OM19R against *Escherichia coli* ATCC25922. The red dotted box is the sequence of the parent peptide OM19R.

**Figure 2 antibiotics-10-01465-f002:**
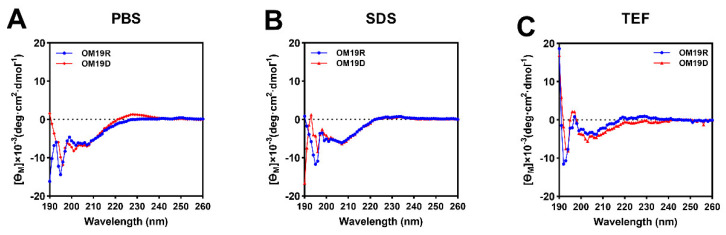
The circular dichroism (CD) spectra of OM19D and OM19R were dissolved in (**A**) 10 mM phosphate-buffered saline (PBS), (**B**) 30 mM sodium dodecyl sulfate (SDS), and (**C**) 50% trifluoroethanol (TFE). The average value after three scans of every sample is shown. The CD spectrum of the buffer was subtracted.

**Figure 3 antibiotics-10-01465-f003:**
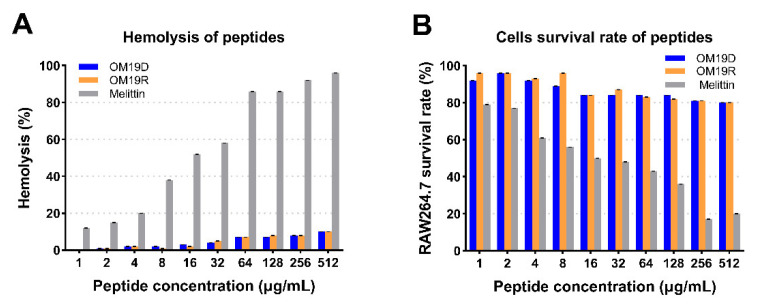
(**A**) Cytotoxicity of peptides against RAW 264.7 cells. (**B**) Hemolytic activity of peptides against rabbit red blood cells. The peptide concentration range is 1–512 µg/mL, melittin and OM19R performed as controls.

**Figure 4 antibiotics-10-01465-f004:**
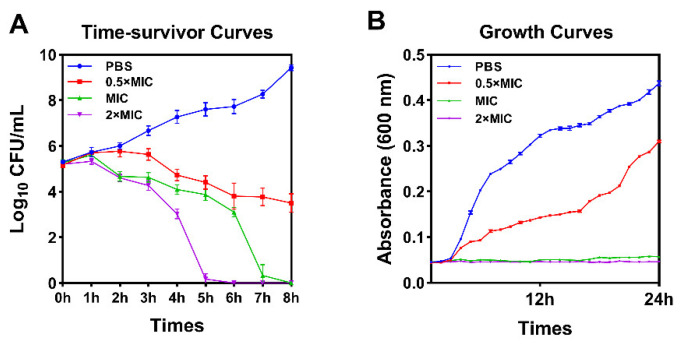
Time-killing curve (**A**) and Growth curves (**B**) for OM19D against *Escherichia coli* ATCC25922. Cells suspensions were incubated with a final concentration of peptides of 0.5 × MIC, MIC, 2 × MIC and controls were performed without the peptide.

**Figure 5 antibiotics-10-01465-f005:**
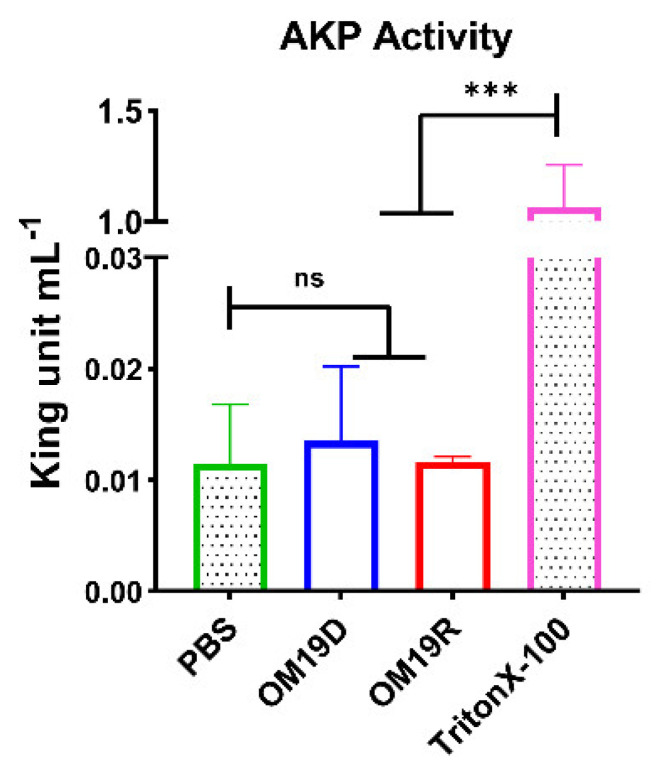
The AKP activity of *Escherichia coli* ATCC25922 was treated with MIC of OM19D and OM19R, the control group was bacteria treated without any peptide. All data are presented as mean ± SD and the significances were determined by nonparametric one-way ANOVA (*** *p* < 0.001, ns represents insignificant).

**Figure 6 antibiotics-10-01465-f006:**
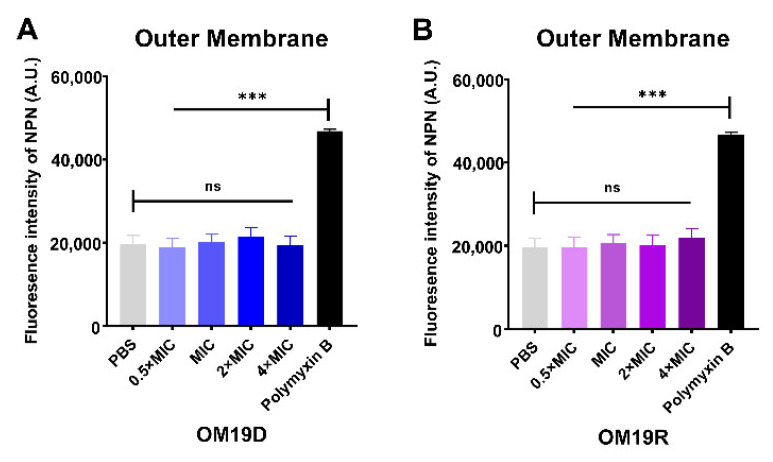
Outer membrane permeability of OM19D (**A**) and OM19R (**B**) at the concentrations from 0.5 × MIC to 2 × MIC. Inner membrane permeability of OM19D (**C**) and OM19R (**D**) at the concentrations from 0.5 × MIC to 2 × MIC. All data are presented as mean ± SD and the significances were determined by nonparametric one-way ANOVA *** *p* < 0.001, ns represents insignificant).

**Figure 7 antibiotics-10-01465-f007:**
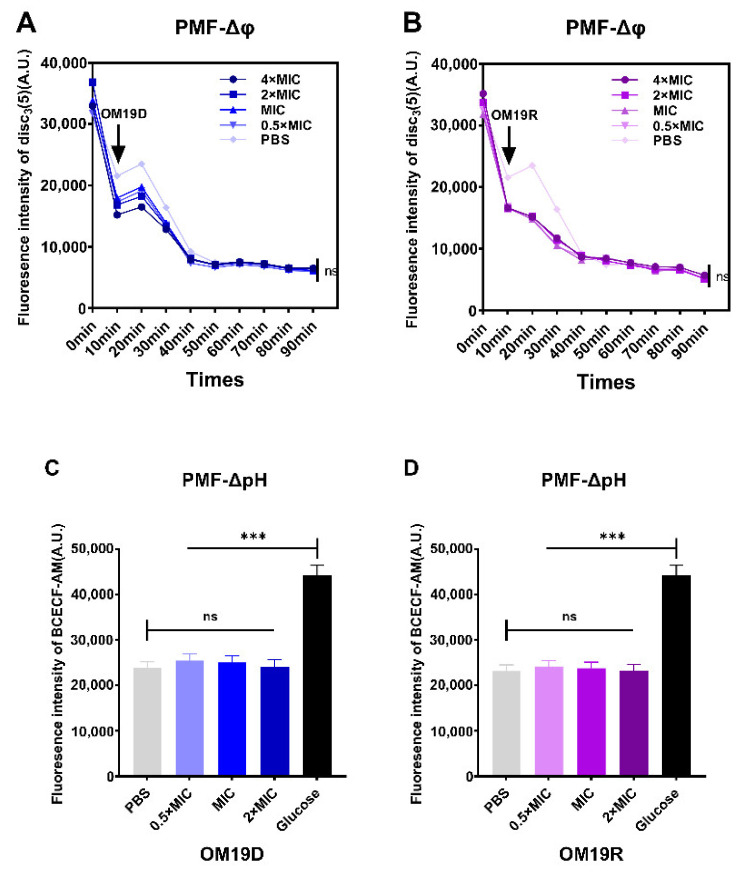
Effects of OM19D (**A**) and OM19R (**B**) on bacterial membrane potential (Δφ) in the concentration range of 0.5 × MIC to 4 × MIC, controls were performed without the peptide. Effect of OM19D (**C**) and OM19R (**D**) on bacterial intracellular pH (ΔpH) in the concentration range of 0.5 × MIC to 4 × MIC, PBS and glucose were the control groups. All data are presented as mean ± SD and the significances were determined by nonparametric one-way ANOVA (*** *p* < 0.001, ns represents insignificant).

**Figure 8 antibiotics-10-01465-f008:**
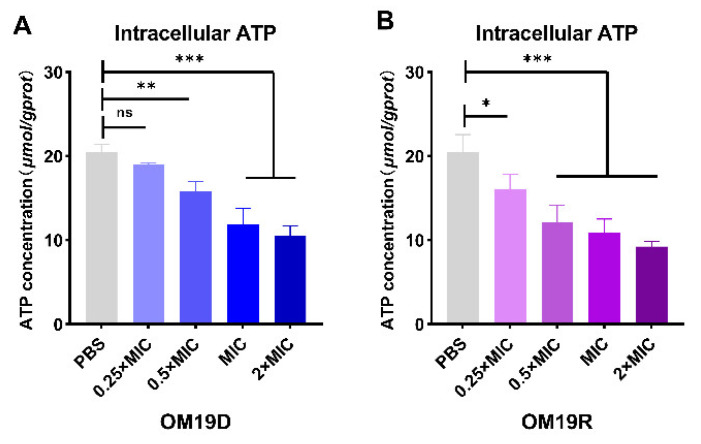
Effects of different concentrations of OM19D (**A**) and OM19R (**B**) on intracellular ATP in *Escherichia coli* ATC25922. All data are presented as mean ± SD and the significances were determined by nonparametric one-way ANOVA (* *p* < 0.05, ** *p* < 0.01, *** *p* < 0.001, ns represents insignificant).

**Figure 9 antibiotics-10-01465-f009:**
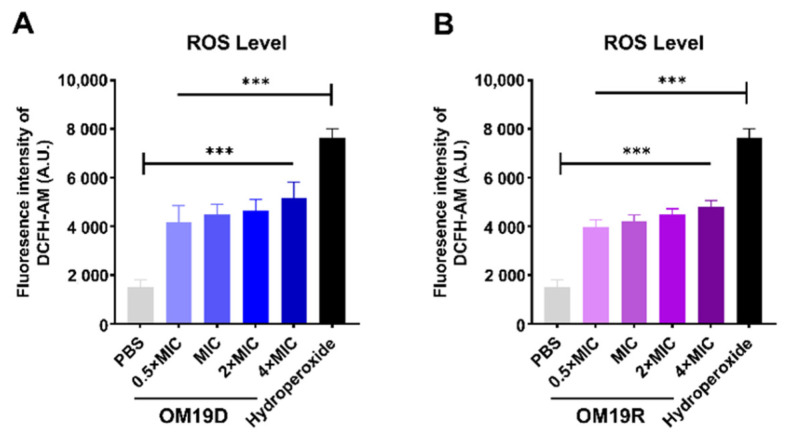
Effects of intracellular reactive oxygen species in *Escherichia coli* ATCC 25922 treated with OM19D (**A**) and OM19R (**B**) at concentrations ranging from 0.5 × MIC to 4 × MIC. The negative control was PBS, and the positive control was hydroperoxide. All data are presented as mean ± SD and the significances were determined by nonparametric one-way ANOVA *** *p* < 0.001.

**Figure 10 antibiotics-10-01465-f010:**
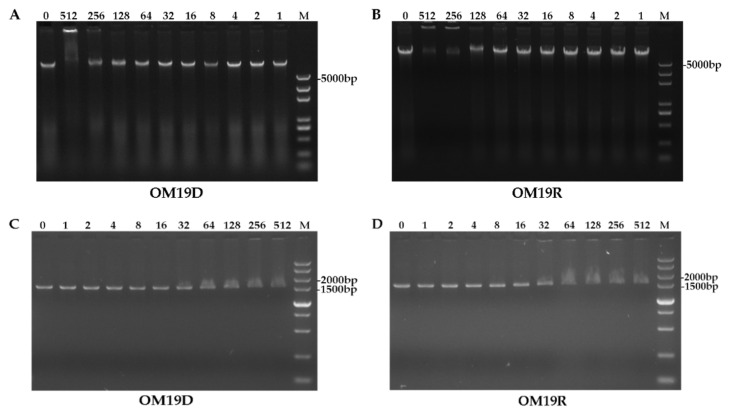
Gel electrophoresis of OM19D (**A**) and OM19R (**B**) from 0–512 μg/mL binding to genomic DNA of *Escherichia coli* ATCC25922, Gel electrophoresis of OM19D (**C**) and OM19R (**D**) from 0–512 μg/mL binding to small DNA fragments of *Escherichia coli* ATCC25922, the letter M stands for individual gene markers, and the number stands for concentration values.

**Figure 11 antibiotics-10-01465-f011:**
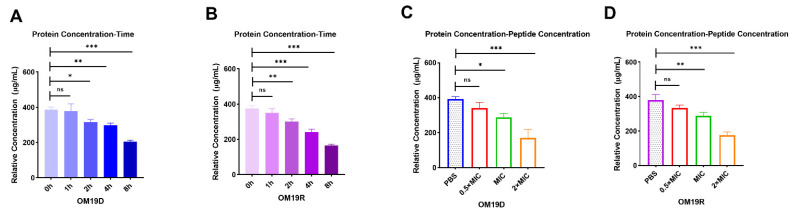
The trend of intracellular protein concentration of bacteria treated with 0.5 × MIC of OM19D (**A**) and OM19R (**B**) followed by time. The trend of intracellular protein concentration of bacteria followed the concentration of OM19D (**C**) and OM19R (**D**). The concentration of peptides ranged from 0.5 × MIC to 2 × MIC. All data are presented as mean ± SD and the significances were determined by nonparametric one-way ANOVA (* *p* < 0.05, ** *p* < 0.01, *** *p* < 0.001, ns represents insignificant).

**Figure 12 antibiotics-10-01465-f012:**
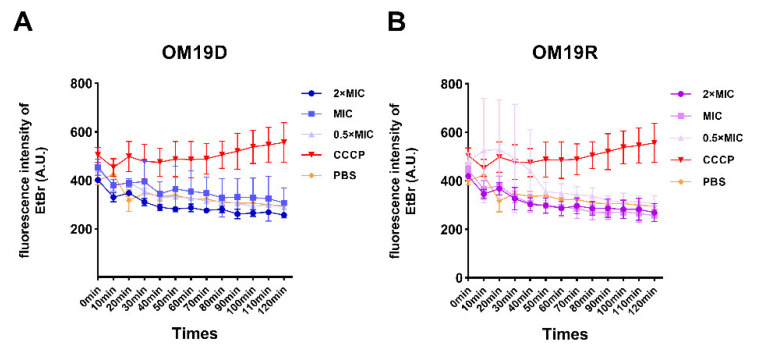
Effects of different concentrations of OM19D (**A**)and OM19R (**B**) on the efflux pump of *Escherichia coli* ATCC 25922.

**Table 1 antibiotics-10-01465-t001:** Peptides and their key physicochemical parameters.

Peptides	Sequence ^1^	Theoretical MW	Measured MW ^2^	Net Charge
OM19D	VDkPPYLPrPrPIrrPGGr-NH_2_	2227.65	2226.63	6
OM19R	VDKPPYLPRPRPIRRPGGR-NH_2_	2227.65	2226.63	6

^1^ Lowercase letters indicate the d-enantiomer. ^2^ Molecular weight (MW) was determined by mass spectrometry.

**Table 2 antibiotics-10-01465-t002:** Minimum inhibitory concentrations (MICs) (µg/mL).

Bacterial Species	OM19D	OM19R
**Sensitive strains**		
*Escherichia coli* ATCC25922	2	1
*Escherichia coli* k88	4	2
*Salmonella enterica* ATCC 14028	4	2
*Salmonella typhimurium* CMCC50115	2	1
*Shigella flexneri* ATCC12022	2	1
*Shigella dysenteriae* CMCC51252	4	2
*Klebsiella pneumoniae* ATCC25955	>128	>128
*Staphylococcus aureus* ATCC25923	>128	>128
**Clinical isolated strains**		
*Escherichia coli* SN5	2	8
*Escherichia coli* S1N1	2	8
*Escherichia coli* w136	4	1
*Escherichia coli* w122	8	2
*Escherichia coli* w123	4	1
*Escherichia coli* w124	4	1
*Escherichia coli* QY	4	4
*Shigella flexneri* QY1	8	8
*Methicillin-resistant Staphylococcus aureus* HP	>128	>128

**Table 3 antibiotics-10-01465-t003:** MIC Values (µg/mL) of peptides against *Escherichia coli* ATCC 25922 in the presence of trypsin.

Peptides	Control ^1^	Trypsin (mg/mL)					Trypsin (10mg/mL)			
		10	5	2.5	1.25	0.625	1 h	2 h	4 h	8 h
OM19D	2	16	16	8	8	4	16	16	16	32
OM19R	1	>128	>128	64	64	64	>128	>128	>128	>128
Melittin	2	>128	>128	64	32	16	>128	>128	>128	>128

^1^ The control MIC values were determined in the absence of proteases.

**Table 4 antibiotics-10-01465-t004:** The MICs (µg/mL) of the designed peptides against *Escherichia coli* ATCC25922 in the presence of physiological salts.

Peptides	Control ^1^	Physiological Salts ^2^					
		NaCl	KCl	NH_4_Cl	MgCl_2_	ZnCl_2_	FeCl_3_
OM19D	2	4	2	2	4	2	4
OM19R	1	4	2	2	4	2	2
Melittin	2	4	2	2	2	2	2

^1^ The control MICs were determined in Mueller–Hinton broth MHB medium without physiological salts. ^2^ The final concentrations of NaCl, KCl, NH_4_Cl, MgCl_2_, CaCl_2_, ZnCl_2_, and FeCl_3_ were 150 mM, 4.5 mM, 6 μM, 1 mM, 2 mM, 8 μM, and 4 μM, respectively.

**Table 5 antibiotics-10-01465-t005:** The MICs (µg/mL) of the designed peptides against *Escherichia coli* ATCC25922 after different temperature and pH treatment.

Peptides	Control ^1^	Temperature				pH			
		0 °C	42 °C	65 °C	100 °C	pH4	pH6	pH8	pH10
OM19D	2	2	2	2	2	2	4	2	4
OM19R	1	1	1	1	1	0.5	0.5	1	1
Melittin	2	2	2	2	2	2	2	2	2

^1^ The control MICs were determined at 37 °C and pH 7.4.

**Table 6 antibiotics-10-01465-t006:** MIC Values (µg/mL) of the Peptides against *Escherichia coli* ATCC 25922 in the presence of Serum and Plasma.

Peptides	Control ^1^	Serum Concentrations (%)				Plasma Concentrations (%)			
		100	50	25	12.5	100	50	25	12.5
OM19D	2	32	32	16	8	64	64	32	16
OM19R	1	32	32	32	16	64	64	64	32
Melittin	2	>128	64	64	32	>128	>128	>128	>128

^1^ The control MIC values were determined in the absence of serum and plasma.

## Data Availability

Not applicable.

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
