# Peer review of "An Antibacterial Peptide with High Resistance to Trypsin Obtained by Substituting d-Amino Acids for Trypsin Cleavage Sites"

_antibiotics, 2021, doi:10.3390/antibiotics10121465_

Round 1

Reviewer 1 Report

In this study Zhao et al., describes how selective substitution of the cationic L-amino acids with D-enantiomers enhance the proteolytic stability of antimicrobial peptide OM19R toward Trypsin. The study shows that selective substitution strategy doesn’t affect the antimicrobial activities or the mechanism of action of this peptide. Further, the authors show that OM19D kills bacteria through non-lytic mechanism by possibly targeting the intracellular components. Over all this is a well constructed study and suitable for the publication in Antibiotics. However, the authors need to address the following concerns before being considered for publication.

Major Comments

  • On the Structure-activity relationships of OM19R, where the authors have used Ala-scanning method to generate a series of analogs of OM19R with Ala substituted at various positions (page 2. line 83). The authors have given the HPLC chromatograms indicating the purity of the peptides used. However, in Fig S2, it is evident that chromatograms b, c, d, m and n are overloaded. It is essential that the authors rerun these peptides at a lower concentration to make sure that the peptides are of the said purity. If the peptides contain considerable amount of impurities, authors need to rerun the MICs and if any any change in the values, results and discussions need to be changed accordingly. Further, it would be ideal that the authors indicate the chromatography conditions used.

  • The authors used BCA assay to quantify the intracellular protein concentration to show that OM19R and OM19D reduce the intracellular protein content. However, the authors used a method that doesn’t consider the total number of the cells at the given time, post treatment with the peptides. From the growth curves and killing-kinetics, it is evident that there is a reduction in the total number of cells as the treatment time progress. Therefore, the authors need to normalize the protein concentration to the number of the cells at a given time point. It is possible that authors are observing a total reduction in biomass. Also, on Fig 11, authors need to indicate the time points for panel C and D (page 9, 10 lines 240-248).

  • The authors used Gel retardation assay using E. coli genomic DNA to study the affinity for DNA. It is ideal to do these retardation assays with small DNA fragments or plasmids, where the changes in the migration can easily be monitored.      

Minor Comments

Page 1, line 18, “trypsin is an insurmountable obstacle for antibacterial peptides” The authors indicate that trypsin is very specific challenge for AMPs. However, in a physiological context trypsin is limited to GI tract and therefore only for orally delivered peptides. The authors need to provide more clarity to this statement in the abstract.

Page 3, line 101: “MALDI-TOF mass spectrometry results are shown in Figure S1” but the spectra and Method given indicate ESI-MS spectra.

Page 3 Table 1: The net charge of the peptide is given as 5, however these are C-terminal amidated peptides. Consequently, there will be a loss of one COOH group. There are total of 6 cationic residues and one Aspartate, which gives a net charge of +5. Since the C-terminal is amidated, the N-terminal amine will increase the charge by +1.

Page 3, line 110: “10Mm PBS” to 10 mM PBS.

Page 4 lines 127-128: The authors have to provide the reference of these strains with respect to their antibiotic resistance profile, if no reference exists, the MICs indicating antibiotic resistance to various classes of drugs should be included as supplementary data.

Page 4 table 2: There are many typos with strain names, authors should correct these strain names. Also, authors must include complete names of the bacteria, “Salmonella H9812” lacks the species name and “MRSA HP” should be spelled out as methicillin resistant S. aureus.

Page 5 Table 3: The results given on the table indicate that though all the Trypsin recognition amino acids are substituted with D-amino acids, there was an 8-16 folds increase in the MIC of OM19D. Interestingly, the method described in the section indicates that precipitation of trypsin (page 12, lines 382-383). It would be interesting to know the loss of activity was actually due to the degradation of peptide, there is a potential possibility that a fraction of peptide is co-precipitated with Trypsin that resulted in an increased MIC.

Page 6 Fig 3: In the figure legend, panels are miss labelled.

Page 6: The authors need to include a positive control such as PB or TritonX-100 in the assay for ALP.

Page 10 line 268: The authors describe the peptide as potential fungicide, however no data on its antifungal activity has been provided in this manuscript.

Page 11 line 313: “random curl” should be rewritten as random coil

Page 16 line 560: “CD chromatogram” should be rewritten as CD spectra.

Page 16: In the conclusion authors indicate that peptide exerts activity by targeting intracellular components, while the same may be true for other Proline rich antimicrobial peptides, there is no conclusive evidence that shows the intracellular localization of the peptide. Therefore, the intracellular targets could only be remaining as a potential possibility.  

Figures in general: The resolutions of these figures need to be improved, in many figures, included texts are pixelated.

Author Response

Thank you for your advice, I have revised the manuscript with reference to your requirements. Please see the attachment.

Reviewer 2 Report

The manuscript entitled “An Antibacterial Peptide with High Resistance to Trypsin Obtained by Substituting D-amino acids for Trypsin Cleavage Sites” describes the use of D-aa (Lysine and Arginine) instead of their respective natural (L-aa) counterparts, which represent trypsin cleavage sites of the antimicrobial peptide OM19R. The authors showed that the aa replacement confers greater resistance to degradation by Trypsin. Also, a detailed comparative analysis between the original peptide OM19R and the modified one, OM19D, demonstrated that the replacement does not affect the structure nor the biological activity of the peptide. The AMP resistance to degradation by proteases is crucial when these molecules are intended for systemic use.

The work is well organized, within the scope of the journal, and should be interesting to the general reader.

I have some comments to be addressed by the authors:

1- Supplementary material shows many MS spectra and (RP?) HPLC profiles when there are only two peptides involved in the study. This isn't very clear. Please provide a proper explanation/caption or remove all unnecessary spectra/chromatographic profiles so the reader can associate one representative MS spectrum + RP(?)HPLC profile with every peptide, OM19R and OM19D.

2- In the supplementary figure, there are spectra composed of multicharged species, however they authors refer to them as MALDI-TOF spectra, which generally yield single charge species. I think these are ESI-MS spectra, please check again the data and change accordingly if there is any mistake, also in the text of the manuscript.

3-The figures related to the HPLC runs should not be named as “HPLC spectra” but as HPLC chromatographic profiles. Also, the authors have to mention what HPLC technique was used. Some readers may suppose it was RPC18-HPLC which is the standard technique for peptides. Anyway, this has to be specified regardless of the technique used.

Author Response

(The authors gave the same response as above.)

Round 2

Reviewer 1 Report

The authors have addressed most of my concerns, and the revisions improved the over all quality of the manuscript and suitable for publication in  Antiobiotics. However, I suggest the authors to include the number of cells ( can be either CFU/mL or total number of cells) used for the protein estimation either in the method or in results section (Section 2.8.8. Quantification of intracellular protein concentration), as it is essential for the readers to be informed that protein is estimated from the same number of cells across the treatment group. Further, the Y-axis values (Protein concentration) given on  the figure 11 are appeared to be same as the previous version, though the authors claims that values are normalized to the viability count. The authors need to make sure that this is not an error.    

Author Response

Response
According to your suggestion, the method was supplemented to further explain
that the protein was estimated from the same number of cells throughout the treatment group ( page 14, line 427 454). In addition, I set the Y axis in Figure 11 as the relative concentration and re analyzed the data page 9, line 216). This modification is marked in blue.

We hope that the revised manuscript will fit the publishing requirements. If there is still something to be improved, please come out and I will continue to modify it. Thank you very much.